# An autoregulatory cell cycle timer integrates growth and specification in chick wing digit development

Joseph Pickering, Kavitha Chinnaiya, Matthew Towers*

Department of Biomedical Science, University of Sheffield, Sheffield, United Kingdom

**Abstract** A fundamental question is how proliferation and growth are timed during embryogenesis. Although it has been suggested that the cell cycle could be a timer, the underlying mechanisms remain elusive. Here we describe a cell cycle timer that operates in *Sonic hedgehog* (*Shh*)-expressing polarising region cells of the chick wing bud. Our data are consistent with Shh signalling stimulating polarising region cell proliferation via Cyclin D2, and then inhibiting proliferation via a Bmp2-p27$^{kip1}$ pathway. When Shh signalling is blocked, polarising region cells over-proliferate and form an additional digit, which can be prevented by applying Bmp2 or by inhibiting D cyclin activity. In addition, Bmp2 also restores posterior digit identity in the absence of Shh signalling, thus indicating that it specifies antero-posterior (thumb to little finger) positional values. Our results reveal how an autoregulatory cell cycle timer integrates growth and specification and are widely applicable to many tissues.

DOI: https://doi.org/10.7554/eLife.47625.001

## Introduction

How proliferation and growth are controlled during embryonic development is a difficult question to address because these processes often involve the interaction between intrinsic mechanisms and extrinsic signals, as revealed in early studies, in which limb buds were transplanted between differently-sized salamander species (*Twitty and Schwind, 1931*). Although much is known about how proliferation and growth are controlled by extrinsic signals, little is known about how these processes are controlled intrinsically. One mechanism involves changing the level of a factor beyond a threshold at which it can stimulate or inhibit cell cycle entry. For instance, the progressive depletion of DNA replication factors triggers the mid-blastula transition in *Xenopus* embryos after a defined number of cell cycles (*Collart et al., 2013*). In addition, the increase in the levels of a G1-S phase inhibitor causes cultured adult oligodendrocyte progenitor cells to differentiate after a defined duration of proliferation (*Durand and Raff, 2000*).

We previously showed that polarising region cells of the chick wing bud measure their duration of proliferation in a similar manner to that of cultured oligodendrocyte progenitor cells (*Chinnaiya et al., 2014*). The polarising region is an important developmental organiser of the early limb bud that produces the secreted signal, Sonic hedgehog (Shh) (*Riddle et al., 1993*), which is implicated in providing cells with antero-posterior (thumb to little finger) positional values (*Yang et al., 1997*; *Tickle and Towers, 2017*). The grafting of polarising regions from young to older wing buds showed that proliferation rates are intrinsically determined (*Chinnaiya et al., 2014*). However, both polarising region and oligodendrocyte progenitor cells require permissive signals to maintain them in a proliferative state. In the case of the polarising region, this signal is based on Fgfs (Fibroblast growth factors) emanating from the overlying apical ectodermal ridge—a thickening of the epithelium that rims the distal tip of the wing bud (*Niswander et al., 1993*; *Laufer et al.,*

*For correspondence:
m.towers@sheffield.ac.uk

**Competing interests:** The authors declare that no competing interests exist.

*1994*; *Niswander et al., 1994*). Similarly, the application of Fgfs or Pdgf (Platelet-derived growth factor) maintains the proliferation of cultured oligodendrocyte progenitor cells (*Durand et al., 1997*). In addition, both polarising region and oligodendrocyte progenitor cells require an instructive hydrophobic signal. For polarising region cells, this signal is considered to be retinoic acid (*Chinnaiya et al., 2014*), which emanates from the trunk of the embryo (*Mercader et al., 2000*). The growth of the wing bud away from the influence of retinoic acid signalling then starts the polarising region cell cycle timer (*Chinnaiya et al., 2014*). However, in the case of oligodendrocyte progenitor cells, retinoic acid or thyroid hormone is required to stop the cell cycle timer after a defined duration (*Barres et al., 1994*; *Tokumoto et al., 1999*). The timing of oligodendrocyte progenitor cell proliferation involves the cell-intrinsic control of p27$^{kip1}$ (*Durand et al., 1997*; *Durand et al., 1998*), an inhibitor of the D cyclins, which are the rate-limiting effectors of the G1-S phase transition (*Ohtsubo and Roberts, 1993*). Although these important in vitro studies on oligodendrocyte progenitor cells provide an exemplar for understanding the temporal control of proliferation, it is unclear if similar mechanisms function during embryogenesis.

In this paper, we reveal that polarising region and oligodendrocyte progenitor cells use similar molecular mechanisms to control their duration of proliferation. We provide evidence that Shh signalling stimulates the proliferation of polarising region cells via Cyclin D2 and then inhibits proliferation via p27$^{kip1}$. We also demonstrate how this timer restricts posterior digit development in the chick wing. In addition, our data suggest that Bmp2 is an important component of the timer, playing a dual role in preventing digit development, while simultaneously specifying posterior digit identity.

## Results

### Shh regulates cell cycle gene expression in the polarising region

Previously we demonstrated that Shh signalling intrinsically controls the rate and duration of polarising region cell proliferation (*Chinnaiya et al., 2014*; *Pickering and Towers, 2016*). Thus, the transient inhibition of Shh signalling at HH20 (for approximately 72 h) using cyclopamine increases the percentage of polarising region cells in the G1-phase of the cell cycle within 24 h, indicative of a decreased rate of proliferation (*Figure 1* – data from *Chinnaiya et al., 2014*; *Pickering and Towers, 2016*). This is followed by an acute stimulation in the proliferation rate between 24 and 48 h. During this time, the proliferation of distal mesenchyme cells adjacent to the polarising region is unaffected by the loss of Shh signalling, thus indicating specific effects on polarising region cell proliferation (*Pickering and Towers, 2016*). The proliferation rate then diminishes in the posterior mesenchyme following the termination of Shh expression, but it is still more rapid than in untreated wings (*Figure 1*) The over-proliferation of polarising region cells is associated with the formation of an extra posterior digit, so that the normal pattern of digits running in the anterior to posterior sequence of 1-2-3, becomes 1-2-2-2, or 1-2-2-3 (*Pickering and Towers, 2016*).

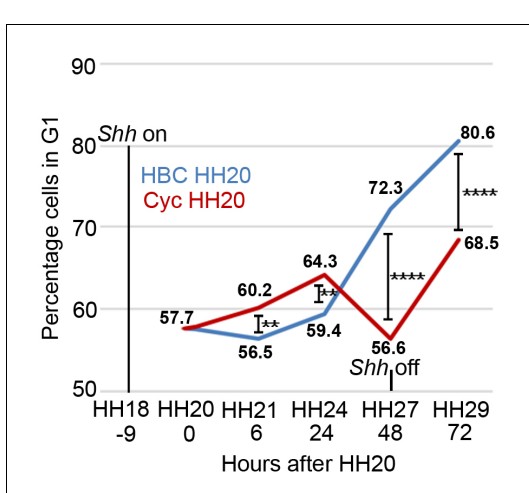

**Figure 1.** Shh signalling influences polarising region proliferation. In normal development (blue line - HBC carrier control) the percentage of polarising region cells in G1-phase of the cell cycle slightly decreases over 6 hr between HH20 and HH21 and then increases throughout wing outgrowth (Note, *Shh* expression terminates at HH27/28 and data and statistics taken from *Chinnaiya et al., 2014*; *Pickering and Towers, 2016*. In all cases, between 10 and 14 polarising regions were dissected and data were obtained from 5 to 10,000 cells). The inhibition of Shh signalling using cyclopamine/HBC at HH20 (red line) increases the percentage of G1-phase cells over 24 hr until HH24, then there is a sharp decrease over the next 24 hr until HH27 when *Shh* expression normally terminates. Over the next 24 hr until HH29, G1-phase cell numbers recover in posterior mesenchyme after *Shh* expression has terminated (unpaired Pearson's $\chi^2$ test **- $p = < 0.01$, and ****- $p = < 0.0001$; see *Chinnaiya et al., 2014*; *Pickering and Towers, 2016*).
DOI: https://doi.org/10.7554/eLife.47625.002

Our previous data provided insights into how polarising region proliferation could be intrinsically controlled. The gene encoding Cyclin D2—an important effector of the G1-S phase transition—is specifically expressed in the polarising region (*Towers et al., 2008*) (*Figure 2a,b* - see expression of *Shh* in polarising region cells - *Figure 2c,d*). Shh signalling is required for the expression of *Cyclin D2* (*Towers et al., 2008*) and this could account for the rapid proliferation of polarising region cells in the early wing bud (*Figure 1*). Therefore, to identify factors intrinsic to the polarising region that could act downstream of Shh signalling to decelerate proliferation at later stages, we determined the expression patterns of the three D cyclin-dependent kinase inhibitors of the *Cip/Kip* group. Neither expression of *p21^{cip1}* (*Figure 2e,f*) nor *p57^{kip2}* (*Figure 2g,h*) is detectable in the chick wing bud. However, *p27^{kip1}* is specifically expressed in the polarising regions of both chick wings and legs over the period that *Shh* is expressed (*Figure 2i–k*), although it is undetectable in the posterior mesenchyme at later stages (*Figure 2l*—note expression in differentiating myogenic cells at HH27/28). In addition, *p27^{kip1}* protein specifically localises to the polarising region of HH24 wing buds (*Figure 2m*).

To examine if Shh signalling is required for *Cyclin D2* and *p27^{kip1}* expression, we systemically applied cyclopamine or control carrier (HBC) to HH20 embryos. *Cyclin D2* expression is undetectable in the polarising region at both 8 and 30 h after cyclopamine is applied (*Figure 3a–d*— both n = 4/4, note, Shh signalling is attenuated within 4 h; *Pickering and Towers, 2016*; and see also *Towers et al., 2008*). However, although expression of *p27^{kip1}* is unaffected at 8 h (*Figure 3e, f*, n = 6/6), it is undetectable at 30 h (*Figure 3g, h*, n = 5/5), coinciding with the time that polarising region cells over-proliferate (*Figure 1*). It should also be noted that cyclopamine does not affect the expression of *Cyclin D1* (*Towers et al., 2008*).

Taken together, these data show that Shh signalling regulates *Cyclin D2* and *p27^{kip1}* expression in a manner that reflects the temporal parameters of polarising region cell proliferation (*Figure 1*).

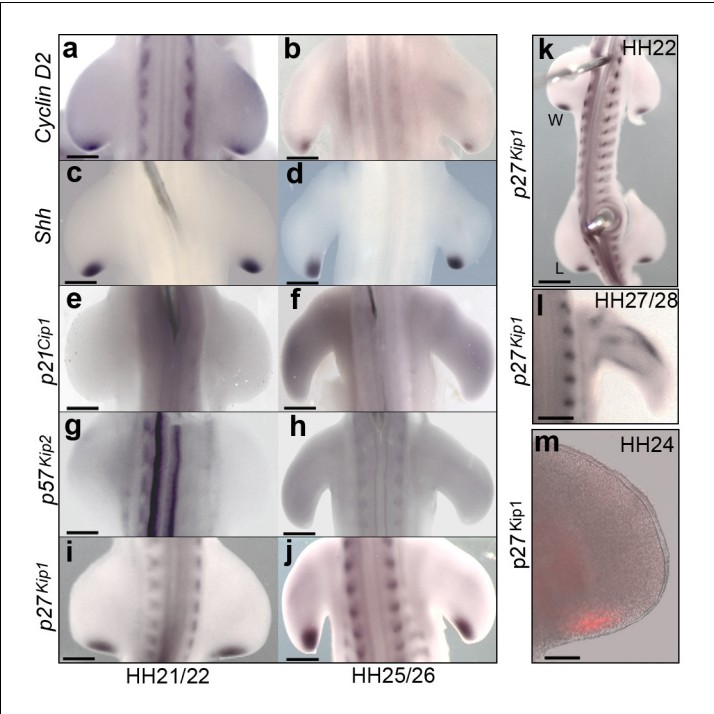

**Figure 2.** The polarising region expresses G1-S phase regulators. Polarising region cells express *Cyclin D2* and *Shh* at HH21/22 (a, c) and at HH25/26 (b, d); but not *p21^{cip1}* and *p57^{kip2}* (e-h). Polarising region cells express *p27^{kip1}* at HH21/22 (i) and HH25/26 (j). *p27^{kip1}* is also expressed in the leg (L) polarising region at HH22, as well as in the wing (W) (k), and in differentiating myogenic cells in the wing at HH27/28 (l). *p27^{kip1}* protein is present in the polarising region at HH24 (m, n = 3/3). Note, gene expression patterns were observed in all embryos analysed (n => 12). Scale bars: a, c, e, g, i, l – 150 µm; b, d, f, h, j – 300 µm; k – 250 µm; - m – 75 µm.
DOI: https://doi.org/10.7554/eLife.47625.003

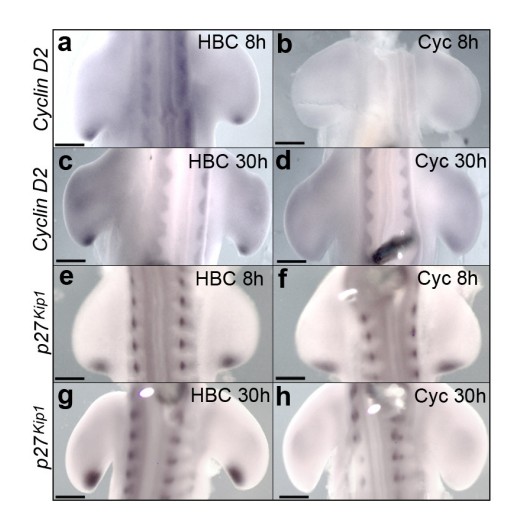

**Figure 3.** Shh signalling regulates *Cyclin D2* and *p27^{kip1}* expression. Application of cyclopamine at HH20 causes loss of *Cyclin D2* expression in the polarising region at 8 hr (a, b, *n* = 4/4 in each experiment) and 30 hr (c, d *n* = 4/4 in each experiment). Although *p27^{kip1}* expression is unaffected at 8 hr (e, f, *n* = 6/6 in each experiment) it is undetectable at 30 hr (g, h, *n* = 5/5 in each experiment). Scale bars: a, b, e, f – 150 μm; c, d, g, h – 300 μm.

DOI: https://doi.org/10.7554/eLife.47625.004

Thus, the requirement of Shh signalling for *Cyclin D2* expression correlates with increased proliferation and the requirement for *p27^{kip1}* expression correlates with decreased proliferation (*Figure 1*).

## D cyclin inhibition prevents polarising region over-proliferation caused by loss of Shh signalling

In wing buds in which Shh signalling is blocked, the loss of *p27^{kip1}* expression could result in the D cyclin-dependent over-proliferation of polarising region cells and additional digit formation (*Figure 1*). To test this hypothesis, we treated chick embryos with PD0332991—a selective inhibitor of all D cyclin/cyclin-dependent kinase complexes (*Toogood et al., 2005*; *Finn et al., 2009*) (note, PD0332991 is marketed as Palbociclib and used to treat breast cancer). In control experiments, the systemic application of PD0332991 to HH20 chick embryos significantly attenuates polarising region proliferation by increasing the number of cells in G1-phase by 31.2% after 6 h (*Figure 4a*, *Figure 4—source data 1A and B*, Pearson's $\chi^2$ test, *p* =<0.0001, *n*=12 polarising regions in each experiment). However, this inhibition is short-lived, as the number of G1-phase cells in PD0332991-treated wings is less than 1% different by 24 h (*Figure 4a*, *Figure 4—source data 1C and D*, Pearson's $\chi^2$ test, *p* => 0.05, *n* = 12 polarising regions in each experiment). Note, for all $\chi^2$ analyses, the percentage of cells in G1-phase in control and experimental samples were compared to the percentage of cells in S/G2 and M phase.

After validating a dose of PD0332991 that is sufficient to inhibit G1-S phase progression in the chick embryo, we asked if this dose of D cyclin/cyclin dependent kinase inhibitor could prevent the over-proliferation of polarising region cells caused by loss of Shh signalling. We systemically applied PD0332991 to control HBC carrier-treated embryos, and also to cyclopamine-treated embryos when polarising region cells begin to over-proliferate at HH24. In both experiments, this causes a significant increase in the number of G1-phase cells after 6 h: a 24.9% increase in HBC-treated wings and a 26.6% increase in cyclopamine-treated wings (*Figure 4b and c*, *Figure 4—source data 1 - source data 1E–H*, Pearson's $\chi^2$ test, *p* =< 0.0001 and *n* = 12 polarising regions in each experiment) and by 24 h at HH27, there is still a significant difference, but the values are much closer: approximately 4% in both cases (*Figure 4b and c*, *Figure 4—source data 1 - source data 1I–L*, Pearson's $\chi^2$ test, *p* =< 0.0001 and *n* = 12 polarising regions in each experiment).

These findings show that D cyclin inhibition can prevent the over-proliferation of polarising region cells caused by the loss of Shh signalling.

## D cyclin inhibition prevents digit formation caused by loss of Shh signalling

To examine the outcome of transient D cyclin inhibition on wing bud growth, we applied PD0332991 at HH24 and then measured the antero-posterior axis across its widest point after 24 h. In control and cyclopamine-treated wing buds, PD0332991 significantly reduces the width of the antero-posterior axis by 17% and 7%, respectively (*Figure 4d–h*, unpaired student's *t*-test *p* =< 0.0001 and *p* =< 0.05, respectively, *n* = 7 wing buds in each case—the apical ectodermal ridge is marked by the expression of *Fgf8*). Note, there is no significant difference between control wings

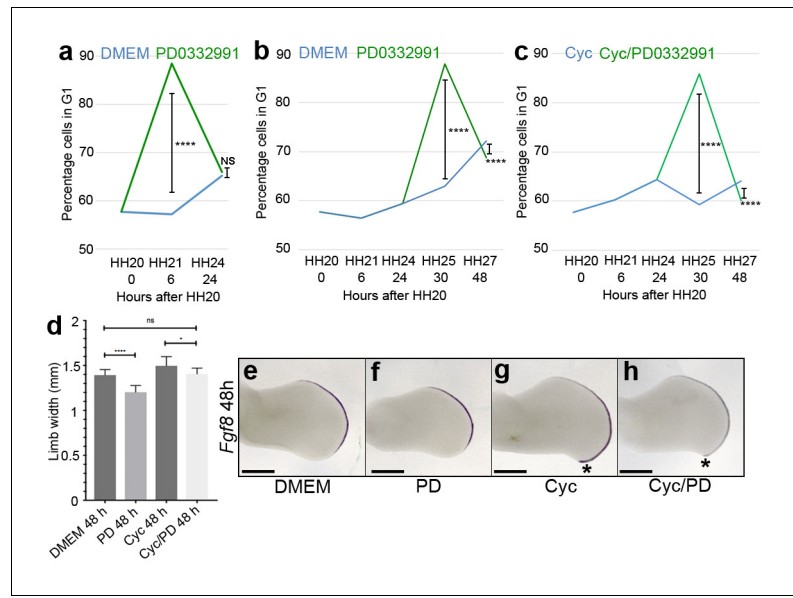

**Figure 4.** D cyclin inhibition prevents polarising region over-proliferation. Application of PD0332991 at HH20 (green line) increases the percentage of polarising region cells in G1-phase to 88.4% after 6 hr compared with 57.2% in DMEM-treated controls (a, blue line - Pearson's $\chi^2$ test - $p < 0.0001$, $n = 12$ polarising regions in both experiments; 11,745 and 11,310 cells analysed, respectively – note HH20 G1-phase data taken from *Chinnaiya et al., 2014*; *Pickering and Towers, 2016*). After 24 hr, G1-phase cell numbers are similar: 65.1% in DMEM-treated polarising regions and 65.9% in PD0332991-treated polarising regions (a - Pearson's $\chi^2$ test – $p > 0.05$ and are not significantly different, $n = 12$ polarising regions in both experiments; 13,062 and 11,833 cells analysed, respectively). Application of PD0332991 at HH24 (green line) to control HBC-treated wing buds (blue line) significantly increases the percentage of polarising region cells in G1-phase to 87.8% (11,932 cells analysed) after 6 hr compared with 62.9% in DMEM/HBC-treated controls (b, blue line - Pearson's $\chi^2$ test - $p < 0.0001$; 5,235 cells analysed $n = 12$ polarising regions in both experiments – note HH20, HH21 and HH24 G1-phase data taken from *Chinnaiya et al., 2014*; *Pickering and Towers, 2016*). After 24 hr, G1-phase cell numbers are still significantly different: 72.2% in DMEM/HBC-treated polarising regions and 68.7% in PD0332991-treated polarising regions (b - Pearson's $\chi^2$ test – $p < 0.0001$, $n = 12$ polarising regions in both experiments; 11,472 and 10,669 cells analysed, respectively). Application of PD0332991 at HH24 to cyclopamine-treated wing buds (green line – note cyclopamine added at HH20) significantly increases the percentage of polarising region cells in G1-phase to 85.9% after 6 hr compared with 59.3% in cyclopamine-treated wing buds (c, blue line - Pearson's $\chi^2$ test - $p < 0.0001$, $n = 12$ polarising regions in both experiments; 10,940 and 11,637 cells analysed, respectively – note HH20, HH21 and HH24 G1-phase data taken from *Chinnaiya et al., 2014*; *Pickering and Towers, 2016*). After 24 hr, G1-phase cell numbers are still significantly different: 64.1% in cyclopamine-treated polarising regions and 60.1% in cyclopamine/PD0332991-treated polarising regions (c - Pearson's $\chi^2$ test – $p < 0.0001$, $n = 12$ polarising regions in both experiments; 11,118 and 10,001 cells analysed, respectively). PD0332991 reduces expansion of the wing bud by 7% compared with DMEM-treated wings, and by 17% compared with cyclopamine-treated wings at 48 hr (d unpaired $t$ test *- $p = {<}0.05$, and ****- $p = {<}0.0001$, $n => 7$ in all cases). Note loss of posterior overgrowth in PD0332991-treated wing buds (asterisks in g and h). Examples of wing buds from which measurements are taken are shown hybridised for *Fgf8* (e-h). Scale bars: 300 µm.

DOI: https://doi.org/10.7554/eLife.47625.005

The following source data is available for figure 4:

**Source data 1.** Flow cytometry graphs for cell cycle analysis.

DOI: https://doi.org/10.7554/eLife.47625.006

and cyclopamine/PD0332991-treated wings (*Figure 4d*). The application of PD0332991 also reduces the pronounced posterior-distal outgrowth seen in cyclopamine-treated wings, which is associated with the formation of an extra digit (*Pickering and Towers, 2016*) (compare asterisks in *Figure 4g and h*).

To assess if the transient inhibition of D cyclin activity at HH24 affects digit patterning, we analysed wings at day 10 of incubation (HH36). Both PD0332991-treated and control DMEM-treated

wings always have the normal pattern of digits (1-2-3 - *Figure 5a, b*, n = 12/12 in both cases). We then determined if PD0332991, when applied at HH24 as polarising region cells over-proliferate, affects the pattern of digits in wings treated with cyclopamine at HH20. The wings of embryos treated with only cyclopamine frequently produce an extra posterior digit—either a digit 2 (14% - *Figure 5c, f*, n = 6/42), or a digit 3 (36% - *Figure 5d, f*, n = 15/42, note that this digit arises from the polarising region; *Pickering and Towers, 2016*). The remaining cyclopamine-treated wings have a normal digit pattern (50%, n = 21/42 - *Figure 5f*). A normal pattern of digits is also found in the wings of approximately half of the embryos (52%) treated systemically with cyclopamine and then PD0332991 (*Figure 5f*, n = 12/23). However, following these treatments, the 1-2-2 digit pattern is found in 40% of wings (*Figure 5e*, f, n = 9/23). The remaining two wings have four digits, suggesting that PD033299 treatment was ineffective (*Figure 5f*, n = 2/23). These results show that 50% of cyclopamine-treated wings have four digits, but only 8% of cyclopamine/PD033299-treated wings have four digits (*Figure 5g*, $\chi^2$ test p =< 0.0001).

Therefore, D cyclin inhibition prevents the formation of an additional digit following the loss of Shh signalling.

## Bmp2 signalling can restore $p27^{kip1}$ expression in the polarising region following loss of Shh signalling

To understand why the inhibition of Shh signalling takes 24 h to attenuate $p27^{kip1}$ expression, we considered the involvement of an intermediate factor. Since Bmp (Bone morphogenetic protein) signalling is known to regulate *Cip/Kip* transcription (*Sharov et al., 2006*; *Nakamura et al., 2003*), we investigated if Bmp2—which is a transcriptional target of Shh signalling in the posterior part of the wing bud (*Yang et al., 1997*)—could control $p27^{kip1}$ expression (Note, *Bmp2* expression is lost within 8 h of cyclopamine treatment; *Scherz et al., 2007*). To examine this, we treated HH20 embryos with cyclopamine, and after 16 h, we implanted beads soaked in recombinant Bmp2 protein into the posterior mesenchyme of right-hand wing buds. In control right-hand wing buds treated with PBS-soaked beads for 8 h, $p27^{kip1}$ expression remains undetectable (*Figure 6a* – n = 3/3). However, in right-hand wing buds treated with Bmp2-soaked beads, $p27^{kip1}$ expression is strongly induced in the polarising region, but not in other regions around the bead (*Figure 6b* – n = 6/6). This result provides evidence that Shh signalling controls $p27^{kip1}$ expression via Bmp2.

## Bmp2 prevents polarising region over-proliferation caused by loss of Shh signalling

To examine if Bmp2 can reverse the over-proliferation of polarising region cells following the loss of Shh signalling, we administered cyclopamine at HH20, and after 6h at HH21, we implanted beads soaked in Bmp2 protein into the posterior mesenchyme of right-hand wing

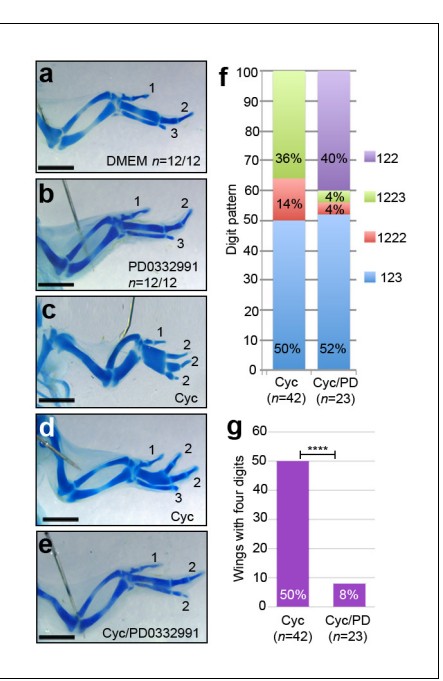

**Figure 5.** D cyclin inhibition prevents digit development caused by loss of Shh signalling. Application of DMEM (**a**) and PD0332991 at HH24 (**b**) does not affect digit patterning (n = 12/12 in both cases). Application of cyclopamine at HH20 results in a 1-2-2-2 digit pattern (**c, f** - n = 6/42, 14%), a 1-2-2-3 pattern (**d, f** - n = 15/42, 36%) and a 1-2-3 pattern (**f**, - n = 21/42, 50%) in day 10 wings. Application of cyclopamine at HH20 and PD0332991 at HH24 results in a 1-2-2 digit pattern (**e, f**, n = 9/23–40%), a 1-2-3 pattern (**f** - n = 12/23, 52%) a 1-2-2-2 pattern and 1-2-2-3 pattern (**f** – both n = 1/23, 4%) in day 10 wings. 50% of cyclopamine-treated wings have four digits, 8% of cyclopamine/PD033299-treated wings have four digits (**g**, $\chi^2$ test p =< 0.0001). Scale bars: 1 mm.
DOI: https://doi.org/10.7554/eLife.47625.007

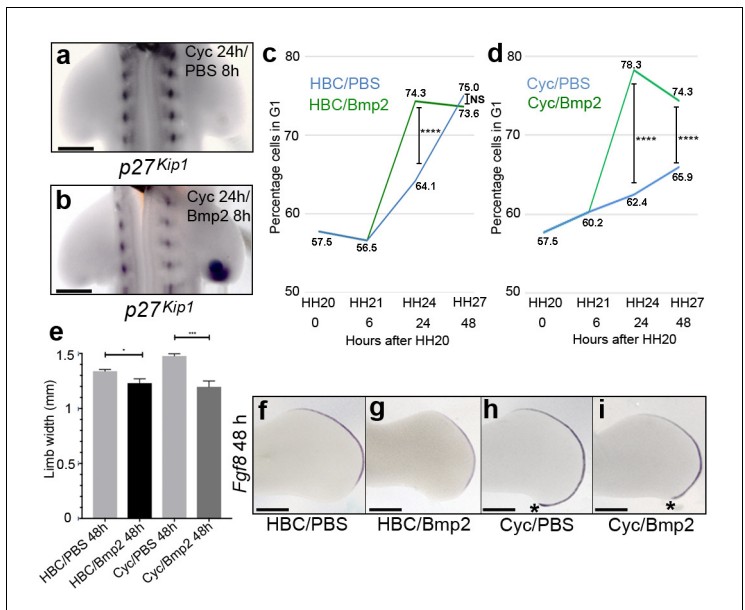

**Figure 6.** Bmp2 regulates *p27^{kip1}* expression and inhibits proliferation. Application of cyclopamine at HH20 and a PBS-soaked bead to the right-hand wing bud 16 hr later causes loss of *p27^{kip1}* expression in the polarising region at 24 hr (**a**, *n* = 3/3). However, a Bmp2-soaked bead induces *p27^{kip1}* in the polarising region (**b**, *n* = 6/6). Application of Bmp2-soaked beads to HH21 wing buds significantly increases the percentage of cells in G1-phase to 74.3% (green line) compared with 64.1% in PBS-soaked bead controls (blue line) at 24 hr (**c**, Pearson's $\chi^2$ test - *p* < 0.0001 – note HBC carrier added also, *n* = 14 polarising regions in both experiments; 7,309 and 9,339 cells analysed, respectively– note HH20, HH21 G1-phase data taken from *Chinnaiya et al., 2014*; *Pickering and Towers, 2016*). After 48 hr, the percentages of cells are very similar (75% and 73.6%, respectively) and not significantly different (**c**, Pearson's $\chi^2$ test – *p* > 0.05; 8,402 and 6,576 cells analysed, respectively). Application of Bmp2-soaked beads to cyclopamine-treated HH20 wing buds (green line) significantly increases the percentage of G1-phase cells to 78.3% compared with 62.4% in PBS-bead/cyclopamine-treated controls (blue line) at 24 hr, and to 74.3% compared with 65.9% at 48 hr. (**d**, *n* = 14 polarising regions in all experiments; 6.924, 9,363, 10,869 and 9,790 cells analysed, respectively - Pearson's $\chi^2$ test - *p* < 0.0001, – note HH20, HH21 G1-phase data taken from *Chinnaiya et al., 2014*; *Pickering and Towers, 2016*). Bmp2-soaked beads reduce expansion of the wing bud by 8.3% compared to PBS bead-treated wings at 24 hr and by 18.9% compared with PBS-soaked bead-cyclopamine-treated wings at 48 hr (**e-i**, unpaired *t* test *- *p* =< 0.05, and ***- *p* =< 0.0005, *n* = 7 in all cases). Note reduced posterior/distal growth in Bmp2-treated wing buds (asterisks in **h** and **i**). Examples of wing buds from which measurements are taken are shown hybridised for *Fgf8* (**f-i**). Scale bars: a, b – 150 µm, e-h - 300 µm.

DOI: https://doi.org/10.7554/eLife.47625.008

The following source data is available for figure 6:

**Source data 1.** Flow cytometry graphs for cell cycle analysis.

DOI: https://doi.org/10.7554/eLife.47625.009

buds. In HBC-treated wing buds, Bmp2 protein beads significantly increase the number of polarising region cells in G1-phase of the cell cycle by 10.2% compared with control PBS-bead treated buds at 24 h (*Figure 6c*, *Figure 6—source data 1A and B*, Pearson's $\chi^2$ test, *p* =< 0.0001 and *n* = 14 polarising regions in each experiment), and by 48 h, there is only a 1.4% change which is not significantly different (*Figure 6c*, *Figure 6—source data 1E and F*, Pearson's Chi-squared test$\chi^2$ test, *p* => 0.05 and *n* = 14 polarising regions in each experiment). Similarly, in cyclopamine-treated wing buds, Bmp2 protein beads significantly increase the number of polarising region cells in G1-phase of the cell cycle by 15.9% compared with control PBS/cyclopamine-treated buds at 24 h (*Figure 6d*, *Figure 6—source data 1C and D*, Pearson's $\chi^2$ test, *p* =< 0.0001 and *n* = 14 polarising regions in each experiment), and there is still a significant 8.4% increase by 48 h (*Figure 6d*, *Figure 6—source data 1G and H*, Pearson's Chi-squared test$\chi^2$ test, *p* =< 0.0001 and *n* = 14 polarising regions in each experiment).

The application of Bmp2 protein also significantly affects the expansion of the wing bud across the antero-posterior axis by 8.3% in control wings buds, and by 18.9% in cyclopamine-treated wing buds (*Figure 6e-i*, unpaired student's *t*-test $p =< 0.05$ and $p =< 0.0001$, respectively, $n = 7$ wing buds in each case—note apical ectodermal ridge is marked by the expression of *Fgf8*). In addition, Bmp2 prevents the pronounced posterior-distal outgrowth seen in cyclopamine-treated wings, which is associated with the formation of an extra digit (*Pickering and Towers, 2016*) (compare asterisks in *Figure 6h and i*).

Taken together, these results suggest that Bmp2 can reverse the effects of the loss of Shh signalling on polarising region cell proliferation.

## Bmp2 prevents digit formation caused by loss of Shh signalling

To determine if the application of Bmp2 to cyclopamine-treated wing buds affects digit patterning, we analysed wings at day 10 of incubation. The application of PBS- or Bmp2-soaked beads to the posterior region of control HH21 wing buds does not affect digit pattern (*Figure 7a,b*, $n = 8/8$ and $n = 10/10$, respectively). Since cyclopamine has identical effects on digit patterning in the left and right-hand wings of approximately 90% of the same embryos (*Pickering and Towers, 2016*), we used this as a control for testing the effects of Bmp2 protein applied to the right-hand wing bud. All of the cyclopamine-treated embryos with three digits in their left-hand wings have a 1-2-3 pattern ($n = 14/14$), and their contralateral wings treated with Bmp2 protein also have a 1-2-3 pattern ($n = 14/14$). In addition, out of the cyclopamine-treated embryos with four digits in their left wings, 47% have the digit pattern of 1-2-2-2 (*Figure 7c, d*, $n = 8/18$) and 53% have the pattern 1-2-2-3 (*Figure 7c,e*, $n = 10/18$). However, all of their contralateral wings in which Bmp2-soaked beads are implanted have three digits, and 94% of these have a digit 3 in a 1-2-3 pattern (*Figure 7c,f*, $n = 17/18$ – note one wing had a 1-2-2 pattern). Therefore, the incidence of a posterior digit 3 increases from 53% in cyclopamine-treated wings to 94% in cyclopamine/Bmp2-treated wings (*Figure 7g*, $\chi^2$ test $p =< 0.0001$).

Taken together, these findings reveal that Bmp2 can prevent additional digit formation and can also specify the digit 3 identity in the absence of Shh signalling.

## Discussion

We have shown that Shh signalling controls an intrinsic programme of chick wing polarising region cell proliferation. Our data suggest that this is mediated via antagonistic regulators of the G1-S phase transition: Cyclin D2 and p27[kip1]. We have provided evidence that Bmp2 signalling functions as part of this timer, both to restrict posterior digit development, and to integrate this process with the specification of antero-posterior positional values.

### An autoregulatory cell cycle timer controls polarising region proliferation

We previously showed that Shh signalling is required for *Cyclin D2* expression in the chick wing polarising region at early stages, which potentially regulates the number of *Shh*-

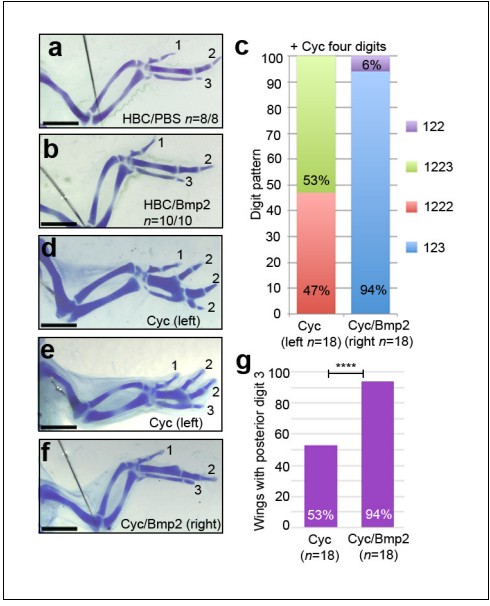

**Figure 7.** Bmp2 inhibits digit development and specifies digit 3. Application of HBC/PBS-soaked beads (**a**) or HBC/Bmp2-soaked beads at HH21 (**b**) do not affect digit patterning ($n = 8/8$ and 10/10, respectively). Wings with four digits following cyclopamine-treatment at HH20 have a 1-2-2-2 digit pattern (**c, d** - 47%, $n = 8/18$) or a 1-2-2-3 pattern (**c, e** - 53%, $n = 10/18$). Application of Bmp2-soaked beads 2 hr later into the right-hand wings buds of the same cyclopamine-treated embryos results in a 1-2-3 digit pattern (**c, f** - 94%, $n = 17/18$) or a 1-2-2 pattern (**c**, 6%, $n = 1/18$). 53% of cyclopamine-treated wings have a posterior digit 3, but 94% of cyclopamine/Bmp2-treated wings have a posterior digit 3 (**g**, $\chi^2$ test $p =< 0.0001$). Scale bars: 1 mm.

DOI: https://doi.org/10.7554/eLife.47625.010

expressing cells (*Towers et al., 2008*) (*Figure 8a*). In addition, *Cyclin D1* is expressed throughout the posterior part of the wing bud, as well as in the polarising region (*Towers et al., 2008*). Since the D cyclins are rate-limiting for G1-S-phase progression (*Ohtsubo and Roberts, 1993*), it is likely that both Cyclins D1 and D2 influence polarising region proliferation, but in being restricted to the polarising region, Cyclin D2 plays a fine-tuning role. We have now demonstrated that Shh signalling is also required for the expression of *p27*$^{kip1}$ in the chick wing polarising region, and we have provided evidence that this occurs via an intermediate signal—Bmp2 (*Figure 8a*). The other two members of the Cip/Kip group of inhibitors, *p21*$^{cip1}$ and *p57*$^{kip2}$, are not expressed in the early chick wing bud, therefore indicating that, unless another inhibitor of the G1-S transition is expressed in the polarising region, p27$^{kip1}$ is responsible for restraining the proliferation of polarising region cells. We propose that Shh signalling activates the Bmp2-p27$^{kip1}$ pathway to prevent the over-proliferation of polarising region cells and the formation of an additional digit (*Figure 8a,b*). The finding that Shh signalling is not required for the maintenance of *Cyclin D1* expression (*Towers et al., 2008*), indicates that the product of this gene is likely to be responsible for the over-proliferation of polarising region cells in the absence of both Cyclin D2 and p27$^{kip1}$ (*Figure 8b*). Therefore, the fact that Shh signalling controls the expression of *Cyclin D2* and *p27*$^{kip1}$ with different temporal dynamics, provides a robust auto-regulatory feedback loop for modulating polarising region proliferation in the chick wing (*Figure 8a*). Our findings are consistent with many studies demonstrating that Shh signalling can either stimulate proliferation via D cyclins (*Kenney and Rowitch, 2000*; *Komada et al., 2013*; *Fink et al., 2018*; *Lobjois et al., 2004*; *Mill et al., 2005*; *Seifert et al., 2010*) or inhibit proliferation via D cyclin inhibitors (*Shkumatava and Neumann, 2005*; *Fu et al., 2017*; *Parathath et al., 2010*). It could be that Shh signalling fulfils both functions in most tissues, but at different times, and this could be the focus of future endeavours.

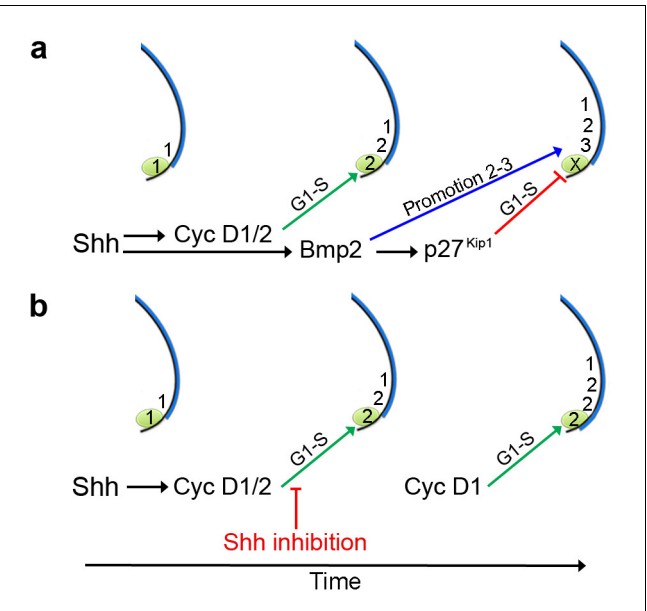

**Figure 8.** An autoregulatory polarising region cell cycle timer. (**a**) Shh signalling is required for *Cyclin D2* expression in the polarising region (green). Bmp2 signalling acts downstream of Shh to control p27$^{kip1}$ that inhibits the D-cyclin-dependent formation of a digit from the polarising region (X), and that also promotes the digit 2 to digit 3 positional value in adjacent cells. Shh signalling also stimulates Cyclin D1-dependent proliferation in the digit-forming field, which is required for the progressive specification of antero-posterior digit positional values - 1, 2 and then 3, over 12 hr. (**b**) Inhibition of Shh signalling during digit specification prevents Bmp2 signalling from promoting the digit 2 to digit 3 positional value in the digit-forming field adjacent to the polarising region. In addition, loss of Bmp2 signalling prevents p27$^{kip1}$ from inhibiting polarising region proliferation and digit development. Since blocking Shh signalling attenuates Cyclin D2 function, Cyclin D1 is likely to be responsible for the over-proliferation of polarising region cells.
DOI: https://doi.org/10.7554/eLife.47625.011

## Bmp2 signalling specifies posterior positional values

The chick wing polarising region produces a concentration gradient of Shh signalling that is considered to directly specify adjacent cells with antero-posterior positional values (*Riddle et al., 1993*). During this process, cells are specified with digit 1 positional values, which are sequentially promoted to digit 2 and then digit 3 positional values over 12 h (*Yang et al., 1997*; *Towers et al., 2011*) (*Figure 8a*). A pattern of anterior digits (1-2-2-2) is often found in wings in which Shh signalling has been blocked during the promotion phase (*Pickering and Towers, 2016*). We have now provided evidence that digits with posterior identity fail to form in such wings, because the loss of Bmp2 signalling prevents the promotion of the digit 2 to the digit 3 positional value (*Figure 8a,b*). Therefore, this finding supports the overlooked suggestion that Bmps function downstream of Shh signalling in the specification of antero-posterior values (*Drossopoulou et al., 2000*). This is an attractive hypothesis because the genes operating downstream of the positional information gradient of Shh signalling in the specification of digit identity have remained elusive (*Tickle and Towers, 2017*). However, the manipulation of Bmp signalling during hand-plate stages in the chick leg can change digit identity (*Dahn and Fallon, 2000*). Therefore, we suggest that Bmp signalling encodes antero-posterior positional values at an early stage of chick limb development, the positional memory of which could be interpreted at later stages of development by the second wave of Bmp signalling in the expanded hand-plate (*Suzuki et al., 2008*).

One of the problems with Bmp2 acting as a long-range signal in digit patterning, is that when it is applied to the anterior margin of the chick wing bud in the form of a recombinant protein, it fails to duplicate the pattern of digits across the antero-posterior axis (*Francis et al., 1994*). However, this can be explained by the requirement for Shh signalling acting as a growth signal to establish the digit-forming field (*Towers et al., 2008*) over which Bmps could then operate (*Drossopoulou et al., 2000*). Another line of evidence against Bmp signalling playing a role in the specification of antero-posterior positional values was the finding that mice lacking Bmp2/4/7 function in their limbs appear to have no overt changes in digit identity (*Bandyopadhyay et al., 2006*). However, we have suggested that the digits of mouse limb (digits 2-5) have similar 'anterior' character in terms of phalanx number and proportions (*Pickering and Towers, 2016*; *Towers, 2018*). Note, that the mouse limb has five digits in the anterior to posterior sequence of 1, 2, 3, 4, and 5. One possibility is that the Bmp2-p27$^{kip1}$ pathway is not as active in mammalian limbs as it is in avian wings, and this could allow the polarising region to produce two digits (*Harfe et al., 2004*). In addition, diminished Bmp2 signalling might only be sufficient to specify anterior positional values, thus resulting in a cryptic '1-2-2-2-2' digit pattern in mammalian limbs. A model such as this could provide an explanation for the digit patterns obtained when Shh function is removed at different time-points during mouse limb development (*Zhu et al., 2008*). We have previously suggested that the digit patterns interpreted as 1-2-4 and 1-2-4-5 (*Zhu et al., 2008*) could instead be '1-2-2' or '1-2-2-2' patterns, because of the failure of Shh signalling to promote sufficient expansion of the digit-forming field to allow four digits to form with a 'digit 2' character (*Pickering and Towers, 2016*; *Tickle and Towers, 2017*). Therefore, we conclude that Shh functions primarily as a mitogen, and that Bmps function downstream of Shh signalling in specifying antero-posterior positional values in the chick wing.

## Implications of the autoregulatory polarising region cell cycle timer

Cell cycle timers have been predicted in several developmental contexts (*Newport and Kirschner, 1982*; *Durand and Raff, 2000*; *Burton et al., 1999*; *Primmett et al., 1989*; *Lewis, 1975*). Recently, we provided evidence that a progressive increase in Bmp signalling is responsible for the slowing of the cell cycle (*Pickering et al., 2018*), which is linked to the acquisition of proximo-distal positional values in the distal mesenchyme of the chick wing bud (humerus to digit tips) (*Saiz-Lopez et al., 2015*). It is tempting to speculate that this could involve a D cyclin-dependent kinase inhibitor acting downstream of Bmp signalling, although it is unlikely to be p27$^{kip1}$, which is only expressed in myogenic cells at later stages of chick wing development. However, important work showed that the intrinsic control of p27$^{kip1}$ is required for the timely differentiation of cultured adult oligodendrocyte progenitor cells after a defined duration of proliferation (*Durand et al., 1997*; *Durand et al., 1998*). Our work extends these earlier studies by providing evidence for a role of p27$^{kip1}$ in cell cycle timing during embryogenesis.

It is of note that mice lacking the function of p27$^{kip1}$ display postnatal overgrowth, which is consistent with cells in various tissues failing to restrain proliferation (*Fero et al., 1996*; *Kiyokawa et al., 1996*; *Nakayama et al., 1996*). However, human overgrowth syndromes involving p27$^{KIP1}$ are rare (*Grey et al., 2013*), and growth syndromes involving loss-of-function mutations in *D CYCLINS* have not been reported. In addition, neither *p27$^{kip1}$* (*Fero et al., 1996*; *Kiyokawa et al., 1996*; *Nakayama et al., 1996*) or *cyclin D2* mice (*Huard et al., 1999*) have been reported to have limb patterning defects, which could be considered inconsistent with the data presented in this report. This could reflect differences between how chick and mice limbs develop. Alternatively, it is likely that robust mechanisms compensate for the germline deletion of individual cell cycle components. For instance, mice lacking all three D cyclins progress to quite late stages of embryogenesis, because other cyclins normally associated with different cell cycle phases replace their functions (*Kozar et al., 2004*). On the other hand, a role for CYCLIN D2 has been predicted in human limb development from the analyses of gain-of-function mutations, which might not be readily compensated for by other cell cycle regulators (*Mirzaa et al., 2014*). Thus, de novo mutations in *CYCLIN D2*—that produce a stable form of the protein—have been identified in individuals who have a rare overgrowth disorder named Megalencephaly-polymicrogyria-polydactyly-hydrocephalus syndrome (MPPH) (*Mirzaa et al., 2014*). Interestingly, individuals with MPPH syndrome frequently produce an extra posterior digit in both their upper and lower limbs, which in light of our data, could suggest that the stable form of CYCLIN D2 has overridden the SHH-BMP2-p27$^{KIP1}$ pathway (*Figure 8a,b*). Therefore, we suggest that the patterning and growth defects found in individuals with MPPH, and related overgrowth syndromes (*Mirzaa et al., 2013*), are caused by the loss of control over intrinsic cell cycle timing mechanisms that operate in multiple tissues.

## D cyclins and limb growth

Our earlier work (*Towers et al., 2008*) and present data suggest that Shh signalling specifies two growth compartments during chick wing development. This growth does not specify digit number per se, which is determined at later stages by processes considered to involve self-organisation (*Newman and Frisch, 1979*; *Oster et al., 1983*; *Wolpert, 1994*). Thus, long-range Shh signalling by the polarising region promotes expansion of the adjacent anterior digit-forming field, likely via Cyclin D1, to provide sufficient tissue for three digits (*Towers et al., 2008*). Conversely, short-range Shh signalling restricts expansion of the posterior polarising region, likely via Cyclin D2/p27$^{kip1}$, to prevent it producing a digit. It is unclear how these two growth outputs of Shh signalling are mediated simultaneously, but it is likely to involve the spatial distribution of the Gli transcription factors: *Gli1* is expressed in posterior cells, whereas *Gli2* and *Gli3* are expressed more-anteriorly (*Marigo et al., 1996*; *Schweitzer et al., 2000*). The anterior growth response to Shh signalling appears evolutionarily conserved in vertebrates because three digits develop from cells adjacent to the polarising region in the chick wing (*Towers et al., 2011*), chick leg (*Towers et al., 2011*) and mouse limb (*Harfe et al., 2004*). However, the ability of the chick leg and mouse limb polarising region to produce more posterior digits than the chick wing, suggests a high degree of growth plasticity, which could be a consequence of the altered dynamics of the auto-regulatory polarising region cell cycle timer.

Similarities and differences can be drawn between chick wing development and *Drosophila* wing development: Hedgehog signalling emanates from the posterior compartment of the *Drosophila* wing to promote growth of the anterior compartment via the Gli orthologue, Cubitus interruptus (Ci) and its regulation of Cyclin D (*Duman-Scheel et al., 2002*). However, cells in the posterior compartment are unable to respond to Hedgehog signalling because they do not express *Ci* (*Lecuit et al., 1996*). Instead, Hh signals to the posterior compartment through the Bmp2 orthologue, Decapentaplegic, which is expressed at the antero-posterior compartment boundary (*Lecuit et al., 1996*). As well as having only one Gli transcription factor (Ci) (*Orenic et al., 1990*), *Drosophila* has one D cyclin (*Finley et al., 1996*) and one D cyclin inhibitor (Dacapo) (*Lane et al., 1996*). Thus, by extending these gene families, vertebrates appear to have enabled Shh signalling to elicit multiple growth responses in both a spatial and temporal manner.

## Materials and methods

### Chick husbandry

Bovans Brown chicken eggs were incubated and staged according to Hamburger Hamilton (*Hamburger and Hamilton, 1951*) HH19 is incubation day 3, HH24 - day 4, HH27 - day 5, HH29 - day 6, HH30 - day 7, and HH36 is day 10.

### Whole mount in situ hybridisation

Embryos were fixed in 4% PFA overnight at 4°C, dehydrated in methanol overnight at −20°C, rehydrated through a methanol/PBS series, washed in PBS, then treated with proteinase K for 20 mins (10 µg/ml$^{-1}$), washed in PBS, fixed for 30 mins in 4% PFA at room temperature and then prehybridised at 69°C for 2 hr (50% formamide/50% 2x SSC). 1 µg of antisense DIG-labelled mRNA probes were added in 1 ml of hybridisation buffer (50% formamide/50% 2x SSC) at 69°C overnight. Embryos were washed twice in hybridisation buffer, twice in 50:50 hybridisation buffer and MAB buffer, and then twice in MAB buffer, before being transferred to blocking buffer (2% blocking reagent 20% lamb serum in MAB buffer) for 2 hr at room temperature. Embryos were transferred to blocking buffer containing anti-digoxigenin antibody (1:2000) at 4°C overnight, then washed in MAB buffer overnight before being transferred to NTM buffer containing NBT/BCIP and mRNA distribution visualised using a LeicaMZ16F microscope.

### Immunohistochemistry

Embryos were fixed in 4% PFA for 2 hr on ice, washed in PBS and transferred to 30% sucrose overnight at 4°C. Embryos were frozen in OCT mounting medium, fixed to a chuck and cryosectioned immediately. Sections were dried for 2 hr and washed in PBS containing 0.1% triton for 10 mins at room temperature. Sections were blocked in 0.1% Triton, 1–2% hings/serum for 1–1.5 hr at room temperature. Primary antibody solution was added (p27$^{Kip1}$ 1:500 Cell Signalling Technology – 2552) in PBS/0.1% triton/1–2% hings, before coverslips were added and slides placed in a humidified chamber at 4°C for 72 hr. Slides were then washed for 3 × 5 mins in PBS at room temperature. Secondary antibody (1:500 anti-rabbit Alexa Fluor 594 Cell Signalling Technology – 8889S) was added in PBS 0.1% triton/1–2% hings for 1 hr at room temperature. Slides were washed in PBS for 3 × 5 mins and mounted in Vectashield/DAPI medium. Slides were left at 4°C and imaged the following day.

### Flow cytometry

Polarising regions were dissected in ice cold PBS under a LeicaMZ16F microscope using a fine surgical knife, pooled from replicate experiments (12-15), and digested into single cell suspensions with trypsin (0.5%, Gibco) for 30 mins at room temperature. Cells were briefly washed twice in PBS, fixed in 70% ethanol overnight, washed in PBS and re-suspended in PBS containing 0.1% Triton X-100, 50 µg/ml$^{-1}$ of propidium iodide and 50 µg/ml$^{-1}$ of RNase A (Sigma). Dissociated cells were left at room temperature for 20 mins, cell aggregates were removed by filtration and single cells analysed for DNA content with a FACSCalibur flow cytometer and FlowJo software (Tree star Inc). Based on ploidy values cells were assigned in G1, S, or G2/M phases, and this was expressed as a percentage of the total cell number (5,000–12,000 cells in each case). Statistical significance of numbers of cells in different phases of the cell cycle (G1 vs. S, G2 and M) between pools of dissected wing bud polarising region tissue (12–15 in each pool) was determined by Pearson's $\chi^2$ tests to obtain two-tailed *P*-values (significantly different being a *P*-value of less than 0.05 – see *Chinnaiya et al., 2014* – statistical comparisons of cell cycle parameters between the wing buds of the same and different embryos showed that less than 2% difference in percentages of G1 phase cells).

### Shh signalling and D cyclin inhibition

Cyclopamine (Sigma) was suspended in control carrier (45% 2-hydropropyl-β-cyclodextrin in PBS, Sigma, to a concentration of 1 µg/µl) and 4 µl pipetted directly onto embryos over the limb bud, after removal of vitelline membranes. PD0332991 was resuspended in DMEM at a concentration of and 0.1 µg/µl and 20 µl pipetted after removal of vitelline membranes. Note in all cases, control wings were treated with 2-hydropropyl-β-cyclodextrin (HBC) or DMEM only.

## Alcian blue skeletal preparations

Embryos fixed in 90% ethanol for 2 days then transferred to 0.1% alcian blue in 80% ethanol/20% acetic acid for 1 day, before being cleared in 1% KOH.

## Bead implantation

Control affigel blue beads (Biorad) were soaked in PBS/4 mM HCl. Affigel beads were soaked in human Bmp2 protein (0.05 µg/µl[1] - R and D) dissolved in PBS/4 mM HCl. Beads were soaked for 2 hr and implanted into posterior mesenchyme using a sharp needle.

## Acknowledgements

We thank the Wellcome Trust for funding, Cheryll Tickle, Marysia Placzek and Jasmina Tarpy for critical reading, and the University of Sheffield flow cytometry core facility.

## Additional information

### Funding

| Funder | Grant reference number | Author |
|---|---|---|
| Wellcome | 202756/Z/16/Z | Joseph Pickering<br>Kavitha Chinnaiya<br>Matthew Towers |

The funders had no role in study design, data collection and interpretation, or the decision to submit the work for publication.

### Author contributions

Joseph Pickering, Conceptualization, Formal analysis, Investigation, Writing—original draft, Performing and planning most experiments, Analysis and validation of the data, Editing and approving the paper; Kavitha Chinnaiya, Formal analysis, Validation, Investigation, Performing p27 cyclopamine experiments, Analysis and validation of the data, Approving the paper; Matthew Towers, Conceptualization, Data curation, Formal analysis, Supervision, Funding acquisition, Validation, Investigation, Visualization, Writing—original draft, Project administration, Writing—review and editing

### Author ORCIDs

Joseph Pickering  https://orcid.org/0000-0002-5892-5159
Kavitha Chinnaiya  https://orcid.org/0000-0002-3375-420X
Matthew Towers  https://orcid.org/0000-0003-2189-4536

### Decision letter and Author response

Decision letter https://doi.org/10.7554/eLife.47625.014
Author response https://doi.org/10.7554/eLife.47625.015

## Additional files

### Supplementary files

• Transparent reporting form
DOI: https://doi.org/10.7554/eLife.47625.012

### Data availability

Source data are provided for flow cytometry.

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
