## [Decision Letter]

Thank you for sending your article entitled "A Sonic hedgehog-Bmp2-p27 kip1 timer controls chick wing digit identity and number" for peer review at *eLife*. Your article has been evaluated by Marianne Bronner as the Senior Editor and Reviewing Editor, and two reviewers.

Below is summary of the required revisions. In addition, we include the full reviews for your reference.

Essential revisions:

1) The Introduction and the discussion should be substantially expanded. There is a missed opportunity to discuss what is known about the function of Shh to shorten the G1 phase and to accelerate cell proliferation, not just in limbs, but also in other tissues beyond the oligodendrocyte cell culture system that they cite as inspiration for this work. Given the fact that *eLife* has no limitation on word count, the authors could place these findings in a much broader context that could better illustrate why these results represent a substantial advance.

2) Please refrain from over-interpretation. For example, as cyclin/cdk inhibitor arrests cells in G1 with or without cyclopamine, please change the last sentence of that section to a softer interpretation (e.g. "Overproliferation in response to loss of Shh signaling is cyclin/cdk-dependent."). There are similar problems with their interpretations in multiple places like stating that Shh prevents cyclin-dependent phenotypes rather than cyclin inhibition reverses the effects of loss of Shh?

3) There should be statistical analysis of digit number. Since their statement is "these findings demonstrate that Shh signaling prevents the D cyclin-dependent formation of an additional digit", it should be possible to determine whether 21/42 with four digits (regardless of identity) in cyclopamine treated embryos is different from 2/23 with four digits with cyclopamine and PD0332991.

4) Please present the statistics more clearly throughout the text rather than in figure legends and methods. For p values, please clarify how these relate to specific time points and clarify reproducibility of the experiments. For all experimental manipulations, please report the actual numbers of individuals in the affected and unaffected groups.

5) The PD0332991 exposure was systemic, so there wasn't a way to say which ones should have had a cyclopamine phenotype to "rescue". For this reason it makes sense to include a "1-2-3" category. Contrast that with the BMP2 bead implantation experiment that is very local and one-sided, with the contralateral side for comparison, so it makes sense that the 1-2-3 category was removed from the analysis. However, the number of individuals unaffected by cyclopamine should be reported.

6) They provide evidence that Bmp2 is sufficient to inhibit proliferation after cyclopamine treatment and that Bmp2 induces p27 after cyclopamine treatment, but the argument that p27 inhibits proliferation in this context is indirect. Is there another way to halt proliferation in the limb in absence of p27? Even if p27 is sufficient, it may not be necessary. Please rewrite the manuscript to provide a more cautious set of conclusions and acknowledge the various interpretations that cannot be excluded by the data.

*Reviewer #1:*

Here, Pickering and colleagues present the next chapter in a thoughtful and comprehensive series of manuscripts that address longstanding questions about limb outgrowth and its integration with the anterior-posterior specification of digits. Specifically, they build on their 2016 Development paper wherein they demonstrated a surprising role for Shh signaling to restrict distal limb expansion; late exposure to cyclopamine causes digit bifurcations and polydactyly, paradoxical to the early requirement for Shh to expand the limb field. In this manuscript submitted to *eLife*, they take inspiration from in vitro studies of a cell cycle timer in cultured oligodendrocytes to test the hypothesis that a Cyclin D inhibitor, p27kip1, acts downstream of Shh to inhibit Cyclin D-dependent cell cycle progression. They end with a model that ties multiple works together with these new data to suggest Shh initially expands the limb field by activating Cyclin D while at the same time inducing Bmp2 expression. Bmp2 in turn activates p21Kip to 'put the brakes' on distal tissue expansion to establish the appropriate number, and identity, of resulting digits.

The authors first reconfirm a previous observation that Cyclin D2 is expressed in the polarizing region and show that its expression is dependent on Shh signaling. They then show p21Kip1 is also expressed in the polarizing region and downregulated by cylopamine, albeit with a temporal delay relative to Cyclin D2. They next test the hypothesis that CyclinD/cdk activity is required for the increased proliferation after late cyclopamine exposure by showing that a cyclin/cdk inhibitor is sufficient to stall cells in G1 and reduce distal limb width to near control in the presence of cyclopamine. The authors then carry these experiments out longer to test the hypothesis that Shh inhibition of cyclin/cdk prevents formation of extra digits. Indeed, inhibition of cyclin/cdk together with shh inhibition frequently reduces the number of digits from four to three.

Last, the authors investigate the mechanism that results in a delayed responsiveness of p21Kip1 downregulation after cyclopamine exposure. Inspired by the fact that Bmps are known to regulate Cip/Kip expression, and the responsiveness of Bmp2 to cyclopamine, they test the hypothesis that restoring Bmp2 expression by bead implantation after cyclopamine treatment will prevent the loss of p21Kip1 expression. I think that this simple and elegant experiment is the greatest strength of the paper. It allows the authors to present a more complete model of early cyclin D2 activation followed by p21Kip1 expression that is delayed by dependence on the intermediate upregulation of Bmp2.

My major concerns are minimal as I think these data are complete and compelling. However, a few corrections are essential.

1) The final paragraph of the Introduction is abruptly short and should be expanded a little to set the stage for the rest of the manuscript. This is a stylistic criticism.

2) The graphed results of FACS in Figure 1, Figure 4A and B, and Figure 6C have no standard deviations per time point and no indication of how many times these experiments were repeated to show reproducibility.

3) Please calculate and report the p value for Cyc+PD versus DMEM in Figure 4C, since I suspect there is no difference.

*Reviewer #2:*

Description of this manuscript:

In this manuscript, Pickering and colleagues report that Sonic hedgehog (Shh) stimulates proliferation of cells in the polarising region of the chick wing bud by controlling the cell cycle gene Cyclin D2, and at later stages Shh inhibits overproliferation by activating Bmp2-p27kip1. In a series of experimental manipulations of chick wing buds, they show that blocking hedgehog signalling causes polarising region cells to over-proliferate, leading to polydactyly (extra digits), and that this affect can be prevented by either application of Bmp2 protein or by inhibition of cyclin D. Inhibition of Shh by cyclopamine results in loss of Cyclin D2 within 8 hours and p27kip1 within 30 hours, Application of Bmp2 blocks cyclopamine-induced over-proliferation and extra digit formation, and this is associated with a rescue of p27kip1 expression.

Summary:

Overall, I found this to be an interesting study that takes advantage of some unique advantages of the chick limb system and builds on the previous work from the Towers lab. However, there are several major weaknesses, including over-interpretation of the results in several places, holes in the experimental logic, lack of statistical support for conclusions, and deficiencies in scholarship. These issues prevent me from recommending publication in *eLife*. There are also some concerns that this study is an incremental advance over previous work, and therefore it may be of interest to a specialized but not a general readership.

Essential revisions:

1) Previous work, some published more than a decade ago, showed that disruption of Shh signaling causes an increase in the percentage of cells in G1, and that Shh directly regulates entry of cells into S-phase of the cell cycle. Surprisingly, the authors cite only their own studies from 2016. It would enhance the paper and improve the scholarship if the authors could acknowledge previous studies and discuss how their results fit within the context of this previous work. Some of these papers have addressed very similar questions to those examined in this manuscript, both in the limb and in other developing systems. Examples of such publications are Zhu et al., 2008; Seifert et al., 2018; Komada et al., 2013; Fink et al., 2018.

2) They show that application of PD0332991, a D cyclin/cdk inhibitor, increases the percentage of G1 cells in control and cyclopamine-treated limbs. In the Results section, they conclude "These findings suggest that Shh signalling prevents the D cyclin-dependent over-proliferation of polarising region cells." This is an over-interpretation of the results. All that this experiment shows is that the PD0332991 inhibits cell cycle progression, which is the known function of the drug.

3) Results section: The experiments in which PD0332991 and cyclopamine were applied in succession resulted in 1-2-2 digit patterns in 40% of the wings. Normal digit pattern is 1-2-3. 5. Treatment with cyclopamine alone produces either an extra digit 2 (14%), an extra digit 3 (36%), or a normal digit pattern (50% – Figure 5F). The variable effects of cyclopamine alone, and the frequency of digit pattern changes under each condition, calls out for a statistical analysis to determine the significance of the effect. In their interpretation of these results, they state, "these findings demonstrate that Shh signalling prevents the D cyclin-dependent formation of an additional digit". This is an overly strong conclusion based on an overinterpretation of some rather indirect evidence.

4) Results section. "The percentage of polarising region cells in G1-phase of the cell cycle increases to 74.3% in Bmp2-treated wing buds compared with 64.1% in control PBS bead treated buds at 24 h". Is this a significant change? Throughout the paper, the absence of statistical analysis underlines the results.

5) In the cyclopamine + Bmp2 experiments (Results section), it is stated, "of the cyclopamine-treated wings with four digits in their left wings, 47% have the digit pattern of 1-2-2-2…and 53% have the pattern 1-2-2-3…" The problem here is that they report percentages of the…treated wings that developed four digits. How many of the treated embryos developed four digits? It is critical to know the size/percentage/number of individuals in the affected and unaffected groups in order to interpret the distribution of digit patterns within the affected group.

6) If p27kip1 is integral to the limb polarising region timer, then one would expect deletion of p27kip1 enhance to perturb limb patterning. However, this experiment was done in 1996 by Fero et al., Kiyokawa et al., and Nakayama et al., but there were no reported effects on the limbs. How does the finding that p27kip1 is dispensable for limb development affect the conclusion (Discussion section) that "a cell cycle timer involving p27kip1 also operates in vivo to control tissue growth and patterning of a major signalling centre during embryogenesis"?

[Editors' note: further revisions were requested prior to acceptance, as described below.]

Thank you for submitting your article "An autoregulatory cell cycle timer integrates growth and specification in chick wing digit development" for consideration by *eLife*. Your article has been reviewed by Marianne Bronner as the Senior Editor and Reviewing Editor, and two reviewers. The reviewers have opted to remain anonymous.

While the manuscript has been much improved, reviewer 2 continues to worry about statistical analysis. I have included the full reviews and ask you to fix these remaining issues. Although *eLife* does not normally allow for multiple rounds of review, I am willing to make an exception in this case since I think you can deal with this remaining concerns in a relatively short period of time without further experimentation but with better analysis. However, no further rounds of revision will be possible, so I urge you to address in detail each of the points raised by the reviewer below.

*Reviewer #1:*

The resubmitted manuscript by Pickering and colleagues is improved by an expanded Introduction and Discussion section. I also appreciate the revised summary statements of the conclusions that explicitly state the results without over-interpreting the data and the more rigorous presentation of data.

*Reviewer #2:*

1) In the review of the original submission, I noted that this advance was incremental relative to their previous (2014 and 2016) papers on this topic, but my comment was not addressed during the open comment period or in the editor's letter to the authors. Therefore, in fairness to the authors, I considered that point to be moot when reviewing the revised version of the paper and my comments are restricted to the data and interpretation. No further action is required.

2) I have reconsidered my comments on Figure 5. I was critical of the analysis of the variable digit phenotypes that were obtained in the two treatments and I suggested that they examine the statistical significance of the variation. Although the authors report the differences in specific digit patterns, their conclusions are based entirely on whether the limbs have 3 digits or 4 digits; the variation in digit patterns was ignored. I accept that they have chosen to bin the data this way, and the effect of PD0332991 treatment on digit number is clear. If the range of variation produced by cyclopamine is not considered in the interpretation of results, then there is no reason to perform the statistical test that I suggested in my previous reviews. I view this as the authors' prerogative. No further action is required.

3) In Figure 7, the use of 3 graphs to display the results of Bmp treatment is unnecessary. The number of individuals unaffected by cycloplamine (panel C) can be stated in the text and the graph eliminated. To make it more clear that panel h has broken-out the 1223 and 123 categories of panel D, I suggest using the same colors.

4) For the experiments that compare the number of cells in G1 between two treatment groups (such as cyclopamine+Bmp2 vs. cyclopamine+PBS) in Figure 1, Figure 4 and Figure 6, I suggested that they use the Student's t-test, which is the appropriate test to compare a quantitative, single dependent variable (the number of cells in G1) between 2 groups. Instead, the authors used a Pearson's chi-squared test and argued that carrying out a t-test would require additional experiments.

I found it difficult to understand how running a different statistical test on the same data set could require additional experiments. However, after reading this paper several times, I now suspect that the central problem concerns a disparity between the description of the data in the results and the description of the data in the methods. Throughout the text, they refer to changes in "the number of G1-phase cells" or "the number of cells in G1". This implies that they compared the numbers of cells in G1 between groups. However, the methods section describes the use of DNA content to assign cells to G1, S, or G2/M, and these categorical assignments were expressed as a percentage of the total cell number (Materials and methods section). Therefore, they are actually comparing between 2 groups the percentage of cells in 1 of the 3 categories, which is not the same as comparing the number of cells in G1 in 2 treatment groups. In other words, it appears that they are comparing categorical variables rather than a quantitative variable. If this is the case, then they can probably fix the problem by providing a more accurate description of the data in the Results section.

5) Finally, although I have offered recommendations on the appropriate statistical tests, I believe that reviewers should not dictate how authors conduct their studies. The authors have my comments and the reasoning behind my advice, so I will now leave it to the authors to proceed as they see fit.

---

## [Author Response]

Essential revisions:1) The Introduction and the discussion should be substantially expanded. There is a missed opportunity to discuss what is known about the function of Shh to shorten the G1 phase and to accelerate cell proliferation, not just in limbs, but also in other tissues beyond the oligodendrocyte cell culture system that they cite as inspiration for this work. Given the fact that eLife has no limitation on word count, the authors could place these findings in a much broader context that could better illustrate why these results represent a substantial advance.

We overlooked the fact there are no limits on word count and we will expand our Introduction and Discussion section.

2) Please refrain from over-interpretation. For example, as cyclin/cdk inhibitor arrests cells in G1 with or without cyclopamine, please change the last sentence of that section to a softer interpretation (e.g. "Overproliferation in response to loss of Shh signaling is cyclin/cdk-dependent."). There are similar problems with their interpretations in multiple places like stating that Shh prevents cyclin-dependent phenotypes rather than cyclin inhibition reverses the effects of loss of Shh?

We have strived to amend incidences flagged up by the reviewers (and elsewhere) where we have over-interpreted our data.

3) There should be statistical analysis of digit number. Since their statement is "these findings demonstrate that Shh signaling prevents the D cyclin-dependent formation of an additional digit", it should be possible to determine whether 21/42 with four digits (regardless of identity) in cyclopamine treated embryos is different from 2/23 with four digits with cyclopamine and PD0332991.

We have performed chi-squared tests showing that this result is significant. We also performed chi-squared tests for the BMP2 rescue of the cyclopamine-treated wings and this shows that the most-posterior digit 3 identity is a significant result.

4) Please present the statistics more clearly throughout the text rather than in figure legends and methods. For p values, please clarify how these relate to specific time points and clarify reproducibility of the experiments. For all experimental manipulations, please report the actual numbers of individuals in the affected and unaffected groups.

We have documented the p-values for each time-point in both the text and the figure panels and included n-numbers.

5) The PD0332991 exposure was systemic, so there wasn't a way to say which ones should have had a cyclopamine phenotype to "rescue". For this reason it makes sense to include a "1-2-3" category. Contrast that with the BMP2 bead implantation experiment that is very local and one-sided, with the contralateral side for comparison, so it makes sense that the 1-2-3 category was removed from the analysis. However, the number of individuals unaffected by cyclopamine should be reported.

We have included this data for cyclopamine-treated left-hand wings and also the digit patterns in the contralateral BMP-treated wings.

6) They provide evidence that Bmp2 is sufficient to inhibit proliferation after cyclopamine treatment and that Bmp2 induces p27 after cyclopamine treatment, but the argument that p27 inhibits proliferation in this context is indirect. Is there another way to halt proliferation in the limb in absence of p27? Even if p27 is sufficient, it may not be necessary. Please rewrite the manuscript to provide a more cautious set of conclusions and acknowledge the various interpretations that cannot be excluded by the data.

This is a good point and we have strived to avoid over-interpreting our findings.

Reviewer #1:

[…] My major concerns are minimal as I think these data are complete and compelling. However, a few corrections are essential.1) The final paragraph of the Introduction is abruptly short and should be expanded a little to set the stage for the rest of the manuscript. This is a stylistic criticism.

We have amended this.

2) The graphed results of FACS in Figure 1, Figure 4A and B, and Figure 6C have no standard deviations per time point and no indication of how many times these experiments were repeated to show reproducibility.

Figure 1 represents previously published experiments performed in the same manner as the ones in our current paper (Chinnaiya, Tickle and Towers, 2014, Pickering and Towers, 2016). The 0-24 hour time-points in Figure 4B and 0-6 hour time-points in Figures6C also show some of this previous data. This data is not essential for these figures, but we felt that it would help the reader understand the experiments. All other time-points show new data. We have explained this in the revised paper.

In our flow cytometry experiments we analyse between 5,000 and 12,000 cells from 12-15 polarising regions (we need to pool this many polarising regions to get enough cells to produce accurate counts). Therefore, this ensures that our results are based on many replicas of the same experimental manipulation and the large number of cells counted provides very accurate data. For instance, we have shown that G1-phase numbers between left and right wing buds of the same embryos differ by less than 1% (Chinnaiya et al., 2014). In addition, when we carefully stage-match tissue from separate embryos incubated at the same time, we also achieve similar values (Chinnaiya et al., 2014). Thus, when we perform Pearson’s chi-squared analyses, a significantly different result generally equates to a >1-2% difference in G1-phase numbers between the control and experimental tissue (Chinnaiya, Tickle and Towers, 2014). The differences in G1 numbers in the key experiments in this paper range between 10% and 32%, which represent extremely significant changes. In our revised paper we have documented the p-values for each time-point in both the text and the figure panels and added n numbers for the cells and polarising regions analysed.

3) Please calculate and report the p value for Cyc+PD versus DMEM in Figure 4C, since I suspect there is no difference.

We have done this.

Reviewer #2:

[…] Overall, I found this to be an interesting study that takes advantage of some unique advantages of the chick limb system and builds on the previous work from the Towers lab. However, there are several major weaknesses, including over-interpretation of the results in several places, holes in the experimental logic, lack of statistical support for conclusions, and deficiencies in scholarship. These issues prevent me from recommending publication in eLife. There are also some concerns that this study is an incremental advance over previous work, and therefore it may be of interest to a specialized but not a general readership.Essential revisions:1) Previous work, some published more than a decade ago, showed that disruption of Shh signaling causes an increase in the percentage of cells in G1, and that Shh directly regulates entry of cells into S-phase of the cell cycle. Surprisingly, the authors cite only their own studies from 2016. It would enhance the paper and improve the scholarship if the authors could acknowledge previous studies and discuss how their results fit within the context of this previous work. Some of these papers have addressed very similar questions to those examined in this manuscript, both in the limb and in other developing systems. Examples of such publications are Zhu et al., 2008; Seifert et al., 2018; Komada et al., 2013; Fink et al., 2018.

We have broadened our review of the literature and we thank the reviewer for these helpful suggestions. It is widely regarded that Shh is a mitogen and it was difficult to decide which studies to concentrate on. It was also known previously that Shh could inhibit proliferation. We did not intend to focus on our previous work but much of it created the premise of the current paper. However, we have now cited papers that describe links between Shh and D cyclins and their inhibitors. In terms of discussion related to these points, we have suggested that it is important to analyse Shh signalling over time, as it is likely to play both anti-proliferative and proliferative roles in may tissues.

2) They show that application of PD0332991, a D cyclin/cdk inhibitor, increases the percentage of G1 cells in control and cyclopamine-treated limbs. In the Results section, they conclude "These findings suggest that Shh signalling prevents the D cyclin-dependent over-proliferation of polarising region cells." This is an over-interpretation of the results. All that this experiment shows is that the PD0332991 inhibits cell cycle progression, which is the known function of the drug.

This is a good point that we have now amended the Results section. ‘These findings show that D cyclin inhibition can prevent the over-proliferation of polarising region cells caused by the loss of Shh signalling.’

3) Results section: The experiments in which PD0332991 and cyclopamine were applied in succession resulted in 1-2-2 digit patterns in 40% of the wings. Normal digit pattern is 1-2-3. 5. Treatment with cyclopamine alone produces either an extra digit 2 (14%), an extra digit 3 (36%), or a normal digit pattern (50% – Figure 5F). The variable effects of cyclopamine alone, and the frequency of digit pattern changes under each condition, calls out for a statistical analysis to determine the significance of the effect. In their interpretation of these results, they state, "these findings demonstrate that Shh signalling prevents the D cyclin-dependent formation of an additional digit". This is an overly strong conclusion based on an overinterpretation of some rather indirect evidence.

We understand that this point refers to essential requirement outlined by the Editor. We have used Chi squared tests ‘to determine whether 21/42 wings with four digits (regardless of identity) in cyclopamine treated embryos is different from 2/23 wings with four digits in cyclopamine and PD0332991’. We have also softened our conclusion – Line 195-197. ‘Therefore, these findings demonstrate that D cyclin inhibition prevents the formation of an additional digit following the loss of Shh signalling.’

4) Results section. "The percentage of polarising region cells in G1-phase of the cell cycle increases to 74.3% in Bmp2-treated wing buds compared with 64.1% in control PBS bead treated buds at 24 h". Is this a significant change? Throughout the paper, the absence of statistical analysis underlines the results.

Previously, we have shown that G1-phase numbers between left and right wing buds of the same embryos differ by less than 1% (Chinnaiya et al., 2014). In addition, when we carefully stage-match tissue from separate embryos incubated at the same time, we also achieve similar values (Chinnaiya et al., 2014). When we perform Pearson’s chi-squared analyses, a significantly different result generally equates to a >1-2% difference in G1-phase numbers between the control and experimental tissue. The differences in G1 numbers in the key experiments in this paper range between 10% and 32%, which represent extremely significant changes. We have added these p values to both the text and figures.

5) In the cyclopamine + Bmp2 experiments (Results section), it is stated, "of the cyclopamine-treated wings with four digits in their left wings, 47% have the digit pattern of 1-2-2-2…and 53% have the pattern 1-2-2-3…" The problem here is that they report percentages of the…treated wings that developed four digits. How many of the treated embryos developed four digits? It is critical to know the size/percentage/number of individuals in the affected and unaffected groups in order to interpret the distribution of digit patterns within the affected group.

We have included this data.

6) If p27kip1 is integral to the limb polarising region timer, then one would expect deletion of p27kip1 enhance to perturb limb patterning. However, this experiment was done in 1996 by Fero et al., Kiyokawa et al., and Nakayama et al., but there were no reported effects on the limbs. How does the finding that p27kip1 is dispensable for limb development affect the conclusion (Discussion section) that "a cell cycle timer involving p27kip1 also operates in vivo to control tissue growth and patterning of a major signalling centre during embryogenesis"?

There have been many reports of the deletion of cell cycle genes including D cyclins and their inhibitors producing no effects on mouse limb patterning. It is possible that the embryonic cell cycle is extremely robust to such germline knockouts. For instance, mice with deletion of all three D cyclin genes develop to quite late stages of embryogenesis because of compensation by other cyclins. Additionally, Cyclin D2 knockout mice have not been reported to have limb defects. However, the predicted constitutive expression of Cyclin D2 does cause limb defects in MPPH patients, suggesting the cell cycle is unable compensate. Therefore, knockout analyses in mice would have also suggested that Cyclin D2 is not important for limb development, but our data and interpretation of MPPH patients would indicate this is not the case. We have discussed all of these points in relation to p27 and Discussion section. We have also softened our conclusion. ‘Our work extends these earlier studies, by providing evidence for a role of p27^kip1^ in cell cycle timing during embryogenesis’.

[Editors' note: further revisions were requested prior to acceptance, as described below.]

While the manuscript has been much improved, reviewer 2 continues to worry about statistical analysis. I have included the full reviews and ask you to fix these remaining issues. Although eLife does not normally allow for multiple rounds of review, I am willing to make an exception in this case since I think you can deal with this remaining concerns in a relatively short period of time without further experimentation but with better analysis. However, no further rounds of revision will be possible, so I urge you to address in detail each of the points raised by the reviewer below.

Reviewer #2:

1) In the review of the original submission, I noted that this advance was incremental relative to their previous (2014 and 2016) papers on this topic, but my comment was not addressed during the open comment period or in the editor's letter to the authors. Therefore, in fairness to the authors, I considered that point to be moot when reviewing the revised version of the paper and my comments are restricted to the data and interpretation. No further action is required.2) I have reconsidered my comments on Figure 5. I was critical of the analysis of the variable digit phenotypes that were obtained in the two treatments and I suggested that they examine the statistical significance of the variation. Although the authors report the differences in specific digit patterns, their conclusions are based entirely on whether the limbs have 3 digits or 4 digits; the variation in digit patterns was ignored. I accept that they have chosen to bin the data this way, and the effect of PD0332991 treatment on digit number is clear. If the range of variation produced by cyclopamine is not considered in the interpretation of results, then there is no reason to perform the statistical test that I suggested in my previous reviews. I view this as the authors' prerogative. No further action is required.3) In Figure 7, the use of 3 graphs to display the results of Bmp treatment is unnecessary. The number of individuals unaffected by cycloplamine (panel C) can be stated in the text and the graph eliminated. To make it more clear that panel h has broken-out the 1223 and 123 categories of panel D, I suggest using the same colors.

We have eliminated the graph (panel C) and we have changed the colour of panel H (now G) to purple to avoid confusion with the categories.

4) For the experiments that compare the number of cells in G1 between two treatment groups (such as cyclopamine+Bmp2 vs. cyclopamine+PBS) in Figure 1, Figure 4 and Figure 6, I suggested that they use the Student's t-test, which is the appropriate test to compare a quantitative, single dependent variable (the number of cells in G1) between 2 groups. Instead, the authors used a Pearson's chi-squared test and argued that carrying out a t-test would require additional experiments.I found it difficult to understand how running a different statistical test on the same data set could require additional experiments. However, after reading this paper several times, I now suspect that the central problem concerns a disparity between the description of the data in the results and the description of the data in the methods. Throughout the text, they refer to changes in "the number of G1-phase cells" or "the number of cells in G1". This implies that they compared the numbers of cells in G1 between groups. However, the methods section describes the use of DNA content to assign cells to G1, S, or G2/M, and these categorical assignments were expressed as a percentage of the total cell number (Materials and methods section). Therefore, they are actually comparing between 2 groups the percentage of cells in 1 of the 3 categories, which is not the same as comparing the number of cells in G1 in 2 treatment groups. In other words, it appears that they are comparing categorical variables rather than a quantitative variable. If this is the case, then they can probably fix the problem by providing a more accurate description of the data in the Results section.

When we have first described flow cytometry we have added: “Note for all Chi-square analyses the percentage of cells in G1-phase in control and experimental samples were compared to the percentage of cells in S/G2 and M phase”.

5) Finally, although I have offered recommendations on the appropriate statistical tests, I believe that reviewers should not dictate how authors conduct their studies. The authors have my comments and the reasoning behind my advice, so I will now leave it to the authors to proceed as they see fit.

We have continued to use Chi-square tests to maintain consistency with our previous papers.